**EMBO** *reports*

*Scientific Report*

# Opposing kinesin complexes queue at plus tips to ensure microtubule catastrophe at cell ends

John C Meadows[†,*] (ID), Liam J Messin[†], Anton Kamnev, Theresa C Lancaster, Mohan K Balasubramanian, Robert A Cross (ID) & Jonathan BA Millar[**] (ID)

## Abstract

In fission yeast, the lengths of interphase microtubule (iMT) arrays are adapted to cell length to maintain cell polarity and to help centre the nucleus and cell division ring. Here, we show that length regulation of iMTs is dictated by spatially regulated competition between MT-stabilising Tea2/Tip1/Mal3 (Kinesin-7) and MT-destabilising Klp5/Klp6/Mcp1 (Kinesin-8) complexes at iMT plus ends. During MT growth, the Tea2/Tip1/Mal3 complex remains bound to the plus ends of iMT bundles, thereby restricting access to the plus ends by Klp5/Klp6/Mcp1, which accumulate behind it. At cell ends, Klp5/Klp6/Mcp1 invades the space occupied by the Tea2/Tip1/Tea1 kinesin complex triggering its displacement from iMT plus ends and MT catastrophe. These data show that *in vivo*, whilst an iMT length-dependent model for catastrophe factor accumulation has validity, length control of iMTs is an emergent property reflecting spatially regulated competition between distinct kinesin complexes at the MT plus tip.

**Keywords** cytoskeleton; fission yeast; kinesin; microtubules
**Subject Categories** Cell Adhesion, Polarity & Cytoskeleton

## Introduction

Microtubule (MT) length control is important for multiple cellular processes including vesicle transport, mitotic spindle size, chromosome bi-orientation and ciliary function [1–4]. In the fission yeast, *Schizosaccharomyces pombe*, arrays of interphase microtubules (iMT) grow along the long axis of the cell and undergo catastrophe at cell ends. Interaction of iMTs with the cell end cortex is required to maintain cell polarity and correctly position the nucleus and cell division ring [5–7]. The maintenance of cell polarity and positioning of the division ring require transport of the Kelch-repeat protein Tea1 and SH3-domain protein Tea4 to the plus ends of iMTs by association with Tea2 (Kinesin-7) and their deposition at cell ends

following interaction of iMTs with the cell end cortex [6,8–12]. Reconstitution and live-cell imaging experiments reveal that association of Tea2 (Kinesin-7) with the growing plus ends of MTs requires two other components, Mal3 (EB1 homologue) and Tip1 (Clip170 homologue) [13–19]. The presence of the Tea2/Tip1/Mal3 complex, but not its cargo (Tea1, Tea3 and Tea4), at the MT plus end also prevents premature MT catastrophe in the cytoplasm [20], at least in part through a mechanism whereby Mal3 alters the structural architecture of MTs [21–23]. Members of the Kinesin-8 family have attracted particular attention as regulators of MT length because they are both highly processive motors and undergo a conformational switch at the MT plus end that promotes MT disassembly [24–30]. These features have given rise to the "antenna model" for MT length control whereby more Kinesin-8 accumulates at the plus ends of longer MTs, thus increasing the likelihood of catastrophe and MT shrinkage [27–29,31]. Fission yeast contains two Kinesin-8 motors, Klp5 and Klp6, that operate as a functional heterocomplex in interphase. Deletion of either gene causes numerous mitotic defects and overgrowth of interphase MTs, which result in defective nuclear positioning and loss of cell polarity, particularly in longer cells [32–36]. Timely shrinkage of interphase MTs also requires Mcp1, a +TIP that is distantly related to the Ase1/PRC1/MAP65 family of anti-parallel MT binding proteins [37–39]. Although association of Mcp1 with MT plus ends requires Klp6, it remains unclear as to whether Mcp1 functions co-ordinately or in parallel with Klp5/Klp6 to control iMT length [31]. In this study, we employ quantitative fluorescence imaging to propose a novel mechanism of MT length control that is based on spatially regulated competition between opposing kinesin motor complexes for binding sites at the MT plus end.

## Results and Discussion

### Mcp1 is an interphase-specific component of the Klp5/Klp6 complex

We first set out to establish the relation of Mcp1 to the Klp5/Klp6 complex. We find that, like Klp5 and Klp6, the intensity of Mcp1

---

Division of Biomedical Sciences, Centre for Mechanochemical Cell Biology, Warwick Medical School, University of Warwick, Coventry, UK
*Corresponding author. Tel: +44 24 76150414; E-mail: J.C.Meadows@warwick.ac.uk
**Corresponding author. Tel: +44 24 76150414; E-mail: J.Millar@warwick.ac.uk
†These authors contributed equally to this work

increases at iMTs plus ends as they grow and dwell at cell ends and decreases as iMTs undergo shrinkage (Figs 1A and EV1A). Binding of Mcp1 to iMT plus ends requires the motor activity of Klp5/Klp6, indicating that Mcp1 is indeed a cargo of the Klp5/Klp6 complex (Fig EV1B; [38]). Consistently, Mcp1 binds weakly to Klp5 in co-immunoprecipitates from cell extracts (Fig EV1C). Importantly, by monitoring MT dynamics using *GFP-atb2* (α2-tubulin), we find that the dwell time of iMTs at the cell end is extended in the absence of both Klp5 and Klp6 to the same extent as in the absence of Mcp1 and this effect is not additive, indicating that Mcp1 controls destabilisation of iMTs via its association with the Klp5/Klp6 complex (Fig 1B). It should be noted that, as with previous studies, it is not possible to determine whether these fluorescent signals represent individual MTs or bundles of a small number of MTs. Notably though, unlike deletion of either Klp5 or Klp6, loss of Mcp1 does not cause cell polarity defects in elongated *cdc25-22* cells (Fig EV1D: [36]) and does not influence mitotic timing or accuracy of chromosome segregation (Fig EV2A–E). These functions may instead be due to association of Klp5/Klp6 with PP1, a type-1-phosphatase (Dis2) [40,41]. Consistently, Mcp1 is not required for Klp5 and Klp6 to bind the mitotic spindle or kinetochores during mitosis and is not present in the nucleus during mitosis (Fig EV2F and G). These results indicate that Mcp1 is an interphase-specific regulator of Kinesin-8-mediated interphase MT length control in fission yeast, confirming and extending previous observations [31].

## Mcp1 is required for the destabilisation activity of Klp5/Klp6 but not the motility

To further understand the role of Mcp1 in Kinesin-8 function, we examined Klp5/Klp6 motility and accumulation at iMT plus ends in control and *Δmcp1* cells by dual camera live-cell imaging (Fig 1C). We find that Klp5/Klp6 walks along the lattice of iMTs at $134 \pm 28$ nm/s, compared to average iMT growth speed of $69 \pm 20$ nm/s, and accumulates at iMT plus ends, particularly whilst iMTs dwell at the cell end, and then dissociates following MT catastrophe (Fig 1B–E). In fact, Klp5/Klp6 translocates faster ($168 \pm 37$ nm/s) along the lattice of iMTs in the absence of Mcp1, although the growth speed of iMTs is unaffected. These data indicate that Mcp1 is required for the iMT-destabilising function of

Klp5/Klp6, perhaps by promoting its association with curved tubulin, but not for its processive motility. We suggest it is likely that, like KIP3 and KIF18A, the highly processive motility of Klp5/Klp6 depends on MT binding site(s) in the C-terminal tail of those proteins [26,42–44]. To quantify the intensity of Klp5/Klp6 at plus ends, mixed populations of cells lacking Mcp1 and control cells were imaged in the same field of view. Although the absence of Mcp1 does not influence Klp5 or Klp6 stability, as judged by Western blot of whole cell extracts (Fig EV1E), nor its intensity in the nucleus, Klp5/Klp6 accumulated to approximately half the level at iMT plus ends in the absence of Mcp1, even though iMTs dwell for longer at cell ends (Fig 1F). This may reflect a role for Mcp1 in enhancing the affinity of Klp5/Klp6 for the iMT lattice, resulting in additional runs to plus ends, or a role for Mcp1 in either retaining Klp5/Klp6 at plus ends or controlling the oligomeric status of the Klp5/Klp6 complex. Reconstitution and biochemical analysis of the Klp5/Klp6/Mcp1 complex will be needed to distinguish between these possibilities.

## Kinesin-7 and Kinesin-8 complexes antagonistically control microtubule length

We next examined the functional relationship between the Klp5/Klp6/Mcp1 and Tea2/Tip1/Mal3 kinesin complexes. In contrast to Klp5/Klp6/Mcp1, which accumulates in both a MT length- and a dwell time-dependent manner, binding of Tea2 kinesin to plus ends is independent of MT length (Fig 2A) but, like Klp5/Klp6/Mcp1, Tea2 dissociates from plus ends following MT catastrophe. To our surprise, we find that Tea2 remains bound, at the same average intensity, to dwelling MT plus ends for longer in the absence of Klp6 or Mcp1 (Figs 2B and C, and EV3A), indicating that the Klp5/Klp6/Mcp1 complex is required for timely dissociation of the Tea2 complex from plus ends. In the absence of Tea2, iMTs dwell at cell ends for less time than in control cells and undergo frequent MT catastrophe in the cytoplasm before iMTs reach the cell end, as previously observed (Fig 2D–F: [20]). Importantly, we find that deletion of *klp5* and *klp6* or *mcp1* in *Δtea2* cells largely restores MT catastrophe near the cell end and increases MT dwell time, although not quite to that observed in control cells (Fig 2E and F). A similar effect is seen in the absence

**Figure 1. Mcp1 is required for control of interphase microtubule stability by Klp5/Klp6 but not for its motility.**

A   Interphase microtubules (iMTs) (magenta) in fission yeast grow towards the cell end (i), dwell (ii) then shrink (iii).

B   Cells expressing fluorescently tagged α2-tubulin (*atb2*) were imaged every 5 s and the dwell time of ~100 individual iMTs recorded within the final 1.1 µm of the cell for each strain. Red bars signify the mean.

C   Kymographs showing fluorescently tagged Klp5/Klp6 (Kinesin) co-imaged with fluorescently tagged microtubules (MT) in the presence (top panels) and absence (bottom panels) of Mcp1. Banding on MT, caused by unequal incorporation of fluorescence, illustrates the force exerted on the MT by continued growth into the cell cortex. Dashed yellow line indicates the cell end.

D   MT growth speed was calculated from kymographs from control (*n* = 16) and *Δmcp1* cells (*n* = 11), and Klp5/Klp6 walk speed was calculated from multiple individual runs on the MT lattice in control (*n* = 44) and *Δmcp1* cells (*n* = 32).

E   Average intensity of Klp5/Klp6 at the plus ends of iMTs from multiple kymographs of control (*n* = 19) or *Δmcp1* cells (*n* = 14).

F   Mixing experiment to compare fluorescently tagged Klp5/Klp6 levels between cells either expressing (blue, closed arrowheads) or deleted (red, open arrowhead) for Mcp1 distinguished by the absence of fluorescently tagged nuclear envelope protein Cut11 (left panel). Scale bar, 5 µm. Box plot (right panel) shows quantitated fluorescence values for nuclear levels of Klp5/Klp6 in control (*n* = 44) and *Δmcp1* cells (*n* = 45) and at the MT plus end in control (*n* = 64) and *Δmcp1* cells (*n* = 35) prior to shrinkage.

Data information: In (E), data are presented as mean ± s.d. *$P < 0.001$, n.s. (non-significant) $P > 0.05$ (Kolmogorov–Smirnov test). In (D) and (F), boxes show the interquartile range with the median represented between the lower and upper quartiles, and whiskers show the highest and lowest values.
Source data are available online for this figure.

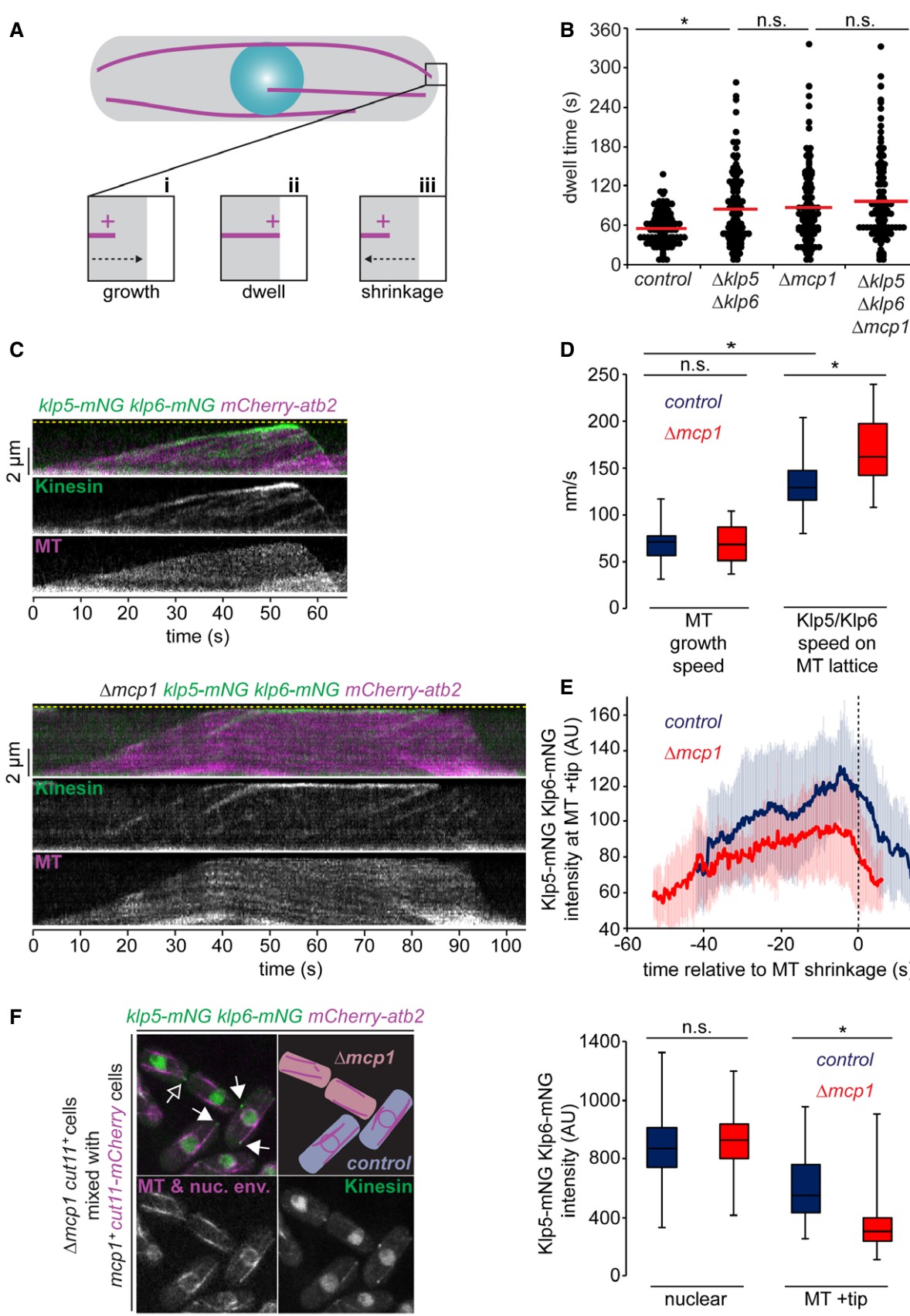

**Figure 1.**

of Tip1 (Fig EV3B). By contrast, MTs dwell only marginally longer at cell ends in Δtea1 cells than the control, and this is exacerbated in Δtea1 Δmcp1 double mutants (Fig EV3B). This suggests that, in the absence of the Tea2/Tip1/Mal3 complex, but not its cargo, Klp5/Klp6/Mcp1 induces premature MT catastrophe in the cytoplasm. Consistently, Klp5/Klp6 accumulates at iMT plus ends in the absence of Tea2, albeit to a lower intensity than in control cells, as iMTs rarely last long enough to reach the cell end (Fig EV3C and D). In the absence of both Tea2 and Mcp1, Klp5/Klp6 accumulates at the plus ends of iMTs that reach the cell end, but at greatly reduced levels compared to control cells (Fig EV3E). In this situation, iMT catastrophe may purely be reliant on the force exerted by interaction of the MT plus end with the cortex at the cell end [27].

### Kinesin-7 restricts access of Kinesin-8 to the microtubule plus end

We next considered how the antagonistic activities of the MT-stabilising Tea2/Tip1/Mal3 and MT-destabilising Klp5/Klp6/Mcp1 complexes are coordinated in space and time. One possibility is that they compete for the same physical space at the iMT plus end. As Klp5/Klp6/Mcp1 concentration increases, it might physically displace the Tea2/Tip1/Mal3 complex from the plus end and induce MT catastrophe. An alternative, but not necessarily exclusive, possibility is that the Tea2/Tip1/Mal3 complex might occlude access by the Klp5/Klp6/Mcp1 complex to the iMT plus end until the iMT encounters the cell end. To test these models, we monitored the relative positions and levels of components of the Tea2/Tip1/Mal3 and Klp5/Klp6/Mcp1 complexes at the iMT plus end. Importantly, dual imaging of *tea2-GFP tip1-tdTomato* cells shows that Tea2 and Tip1 occupy the same region of the MT plus end and remain at the same intensity after MT contact with the cell end cortex, consistent with the fact that these proteins are components of the same complex (Fig 3A and B). By contrast, dual imaging of *klp5-mNG klp6-mNG tip1-tdTomato* cells reveals that Klp5/Klp6 accumulates behind Tip1 as the iMT grows and dwells at the cell end, as judged by maximal intensity measurements of the fluorescence (Figs 4A and B, and EV4, and Movie EV1). Importantly, the Tea2/Tip1/Mal3 and Klp5/Klp6/Mcp1 complexes converge at the plus end approximately

10 s before MT shrinkage is detected. This behaviour is also observed in the absence of Mcp1, although in this case Klp5/Klp6 remains behind the Tea2/Tip1/Mal3 complex at the MT plus end for longer as it dwells at the cell end (Figs 4C and EV5).

These results indicate that the "antenna model" for MT age- and length-dependent accumulation of Kinesin-8 has validity but cannot solely explain the position-dependent interphase function of Klp5/Klp6 in fission yeast. The Klp5/Klp6/Mcp1 complex accumulates at MT plus ends over time, but Kinesin-8 walks only marginally faster than iMT polymerisation, so that its accumulation is largely dependent on MT growth slowing at cell ends rather than on MT length. Importantly, our data indicate that the MT-stabilising Tea2/Tip1/Mal3 kinesin complex binds ahead of the MT-destabilising Klp5/Klp6/Mcp1 complex and restricts its access to the iMT plus end until the MT reaches the cell end (Fig 5). In the absence of both motor complexes, iMTs generally undergo catastrophe near the cell ends, possibly as a result of force-induced suppression of MT polymerisation. Regardless, our data indicate that iMT length control in cells is a complex emergent property reflecting competition and collaboration between multiple factors that influence tubulin dynamics. The earliest events in the sequence that results in MT catastrophe remain unclear. We note that the rate of MT growth slows as the plus ends near the cell end, and this precedes any measurable change in the Tea2 complex signal. One possibility is that slowing down of iMT growth at or very close to cell ends allows Kinesin-8 to build to a critical concentration, which enables it to invade the space occupied by Tea2 and/or displace it from plus ends. In this manner, MT-destabilising Kinesin-8 accumulates at plus ends dependent on MT length and lifetime (as in the antenna model), but its access to the MT plus end is controlled in time and space by an engagement mechanism that slows growth and favours unloading of the Tea2 complex. Another possibility is that Tea2 complex dissociation gates access by the Kinesin-8 complex to the extreme plus end of MTs. What is clear is that competition between stabilising and destabilising kinesin complexes at the plus end provides a spatial control mechanism that overlays the fundamental mechanosensitivity of tubulin dynamics to fine-tune MT length. In the future, it may be valuable to reconstitute both Kinesin-7 and Kinesin-8 motor complexes on dynamic fission yeast MTs as they engage an artificial barrier to formally test this model.

---

**Figure 2.  Tea2 and Klp5/Klp6/Mcp1 antagonistically control microtubule stability.**

A   Log phase cells expressing fluorescently tagged MTs were imaged, and the levels of either fluorescently tagged Klp5/Klp6 or Tea2 at the plus ends of growing MTs were determined. Measurements (*klp5-mNG klp6-mNG*, n = 305; *tea2-GFP*, n = 359) were plotted against microtubule (MT) length and third-order polynomial curves fitted to the data.

B   Kymographs showing fluorescently tagged MTs co-imaged with fluorescently tagged Tea2 kinesin in control cells (top panels) or in the absence of Klp6 (middle panels) or Mcp1 (bottom panels). Dashed yellow line indicates the cell end.

C   Intensity of Tea2 at plus ends quantitated from multiple kymographs of the type in (B) for control (n = 16), Δklp6 (n = 13) or Δmcp1 (n = 14) cells.

D   Kymographs showing fluorescently tagged MTs co-imaged with fluorescently tagged Klp5/Klp6 in control (top panels, repeated from 1C) or Δtea2 cells (bottom panels). Dashed yellow line indicates the cell end.

E   Cells expressing fluorescently tagged tubulin were imaged every 5 s and the dwell time of ~100 individual iMTs for each condition recorded within the final 1.1 μm of the cell. Red bars signify the mean.

F   The strains in (E) were analysed to determine the position of 20 MT shrinkage events relative to the cell end, which are represented both graphically and as a box plot.

Data information: In (C), data are presented as mean ± s.d. *P < 0.001, **P < 0.05, n.s. (non-significant) P > 0.05 (Kolmogorov–Smirnov test). In (F), boxes show the interquartile range with the median represented between the lower and upper quartiles, and whiskers show the highest and lowest values.
Source data are available online for this figure.

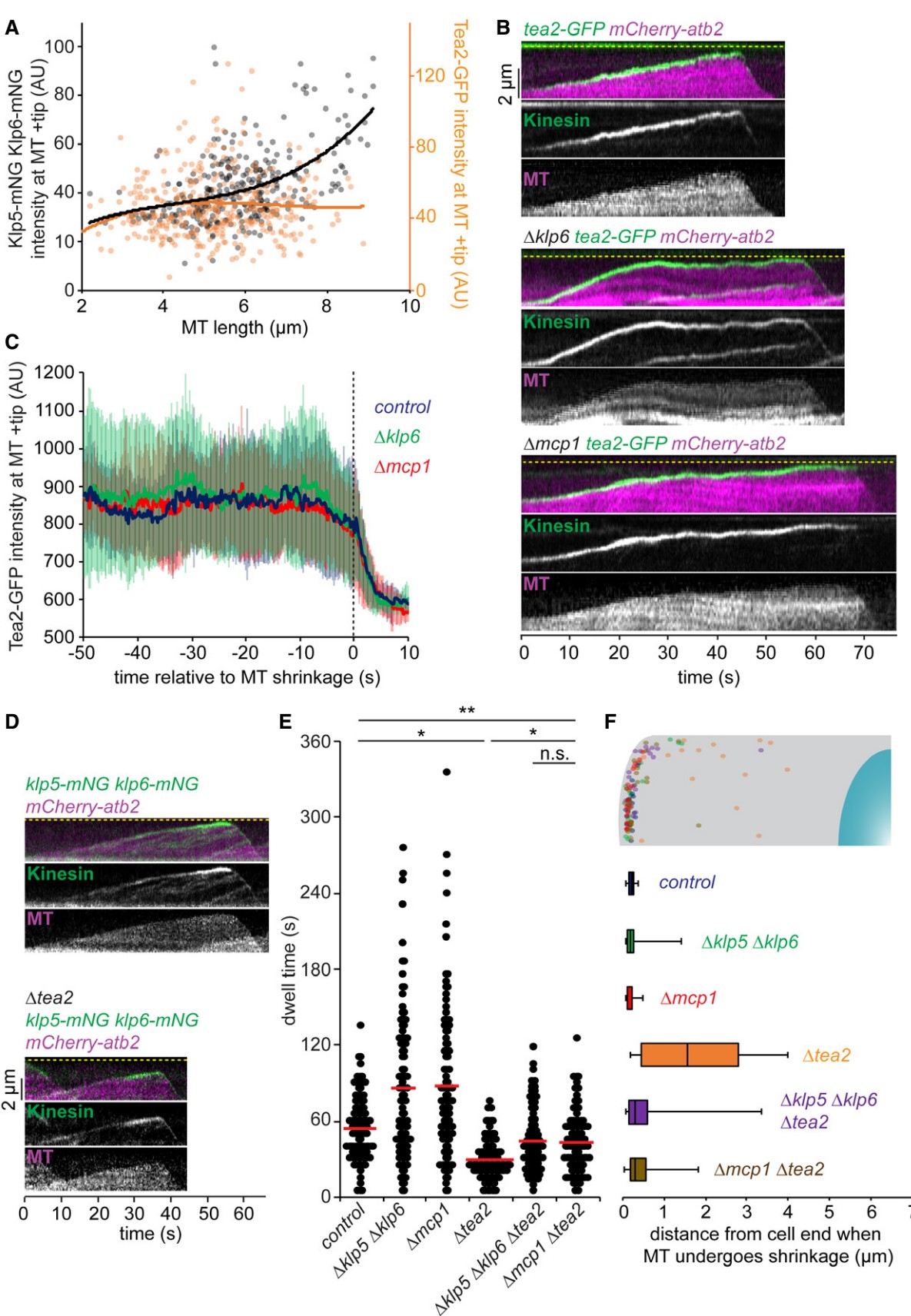

Figure 2.

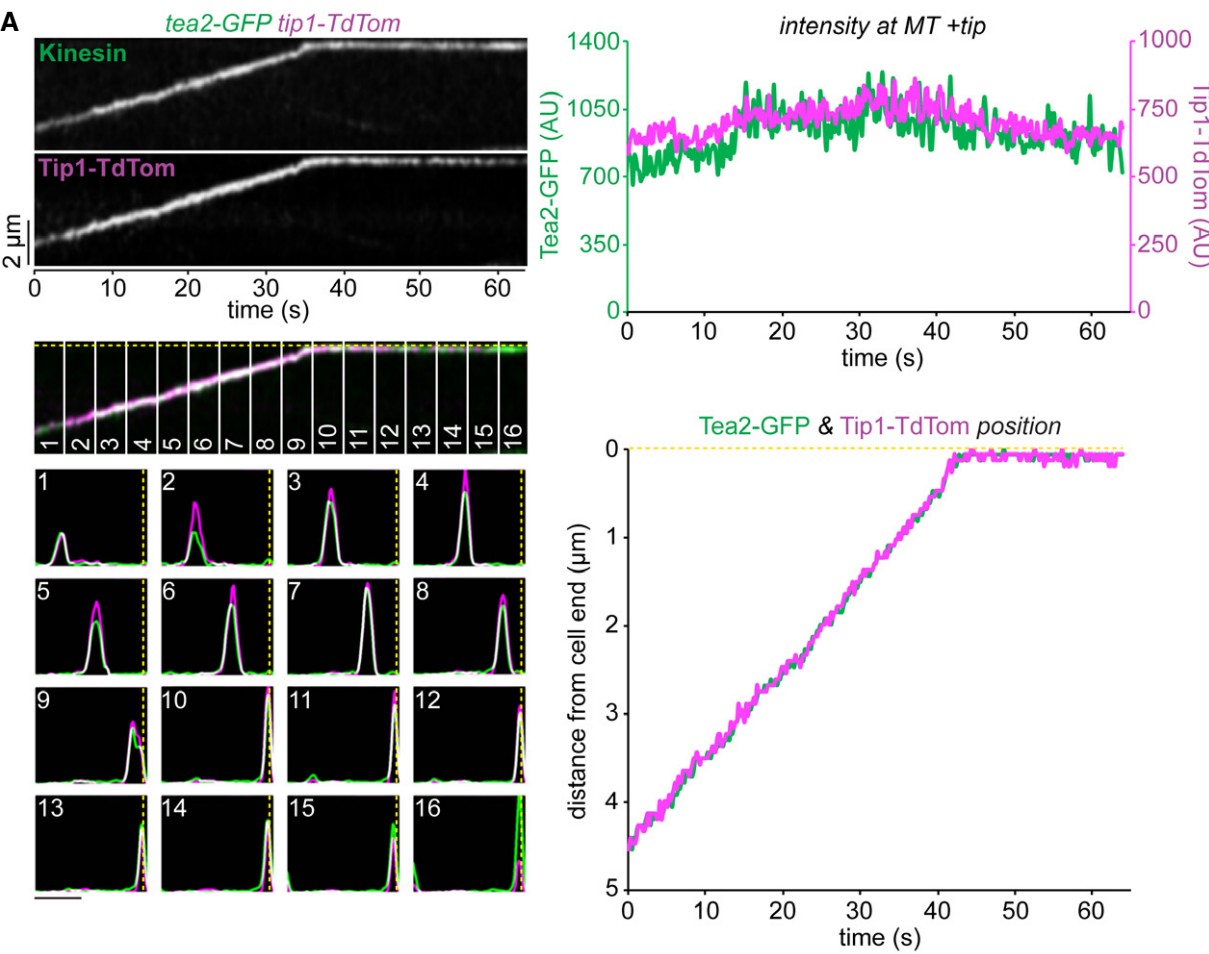

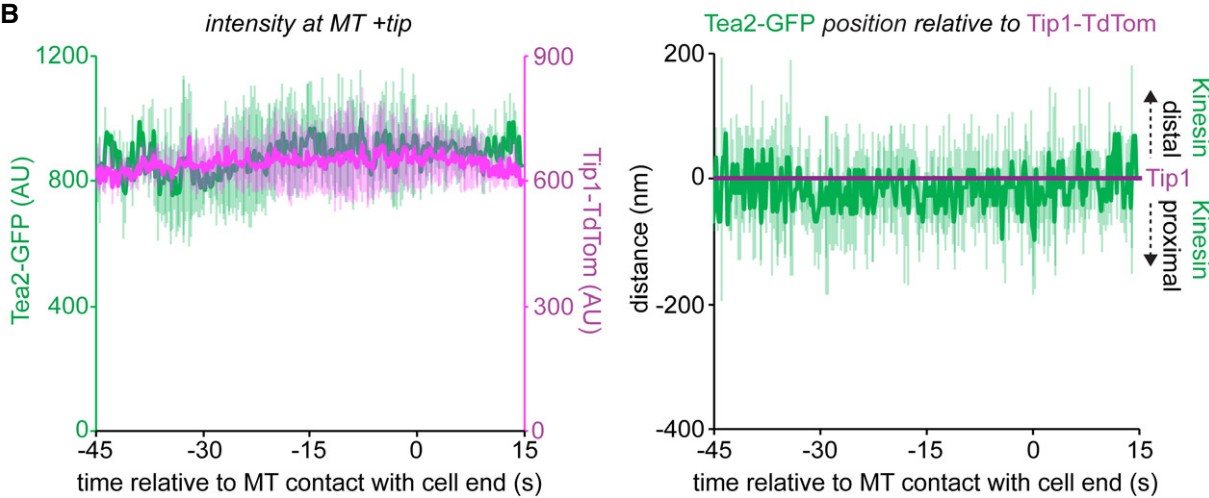

**Figure 3. Tea2 and Tip1 co-localise at the microtubule plus end.**

A Kymograph (top left panels) showing fluorescently tagged Tea2 (Kinesin) co-imaged with fluorescently tagged Tip1 (Tip1-Tdtom). Dashed yellow line indicates the position of the cell end. Plots (bottom left panels) show both the relative fluorescence intensity and position for Tea2 (green) and Tip1 (magenta) corresponding to the numbered sections of the kymograph. Scale bar, 2 μm. Data quantitated from this kymograph by extracting the maximal intensity pixel value at the MT plus end for Tea2 (green) and Tip1 (magenta) over time (top right) and the distance of these pixels from the cell end (bottom right).

B Data quantitated from multiple kymographs (*n* = 5) of the type illustrated in (A) to display Tea2 and Tip1 intensity at the MT plus end (left plot) and the position of Tea2 relative to Tip1 (right panel) and relative to initial MT contact with the cell end. Error bars show standard deviation.

 

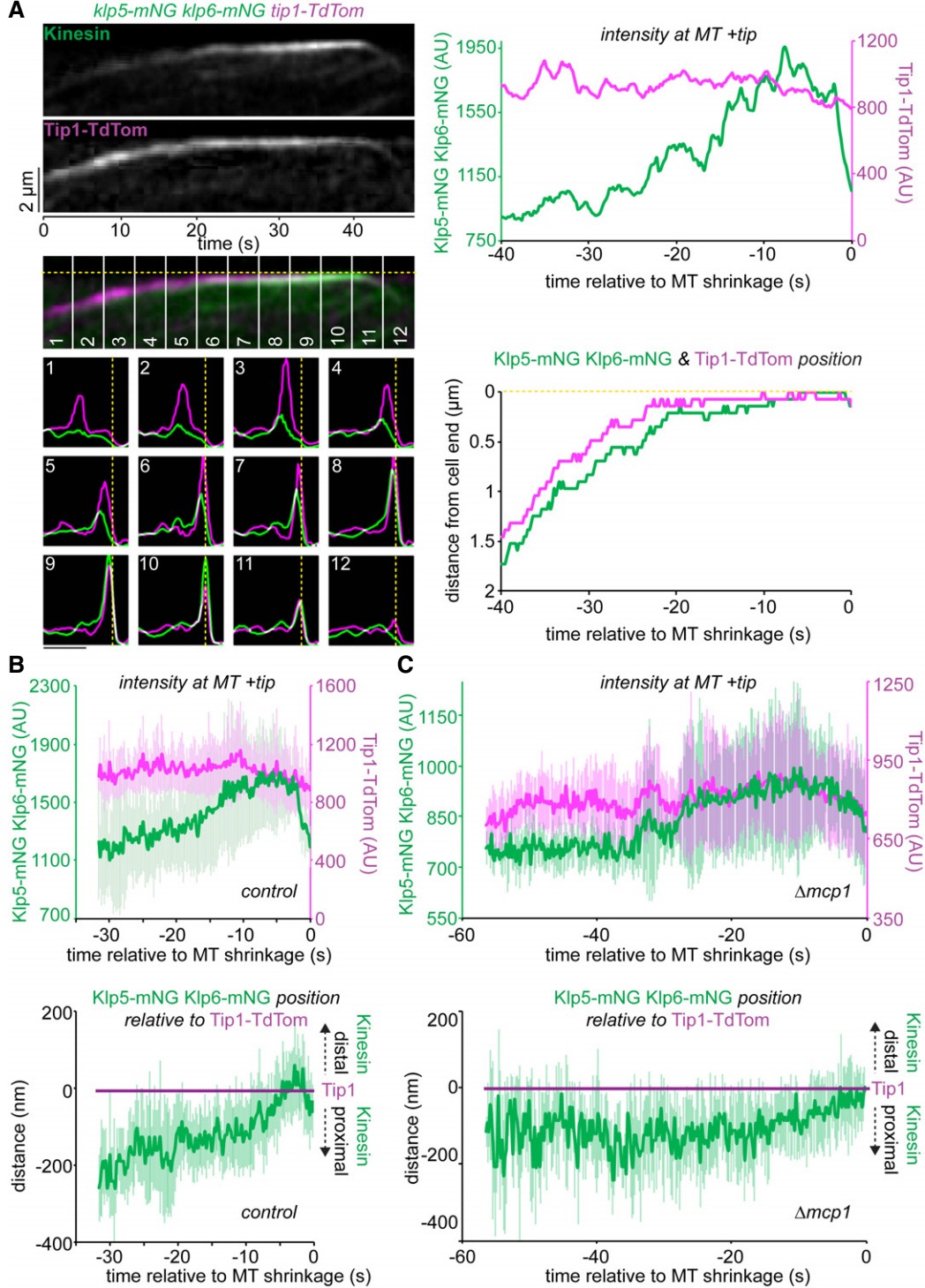

**Figure 4.  Klp5/Klp6 accumulates behind Tea2 at the plus end until just prior to microtubule catastrophe.**

A   Kymograph (top left panels) showing fluorescently tagged Klp5/Klp6 (Kinesin) co-imaged with fluorescently tagged Tip1 (Tip1-Tdtom). Dashed yellow line indicates the cell end. Plots (bottom left panels) show both the fluorescence intensity and position of Klp5/Klp6 (green) and Tip1 (magenta) corresponding to the numbered sections of the kymograph. Scale bar, 2 µm. Intensity of the maximal value pixel at the plus end (top right panel) and its position relative to the cell end (bottom right panel) for both Klp5/Klp6 (green) and Tip1 (magenta) plotted relative to MT shrinkage for this kymograph.

B   Data quantitated from multiple kymographs (*n* = 7) of the type illustrated in (A) and Fig EV4 to display Klp5/Klp6 and Tip1 levels at the MT plus end (top plot) and the position of Klp5/Klp6 relative to Tip1 (bottom plot) before MT shrinkage.

C   Data quantitated from multiple kymographs (*n* = 8) of the type illustrated in Fig EV5 where fluorescently tagged Klp5/Klp6 is co-imaged with fluorescently tagged Tip1 in the absence of Mcp1 and presented as in (B).

Data information: In (B, C), data are presented as mean ± s.d.

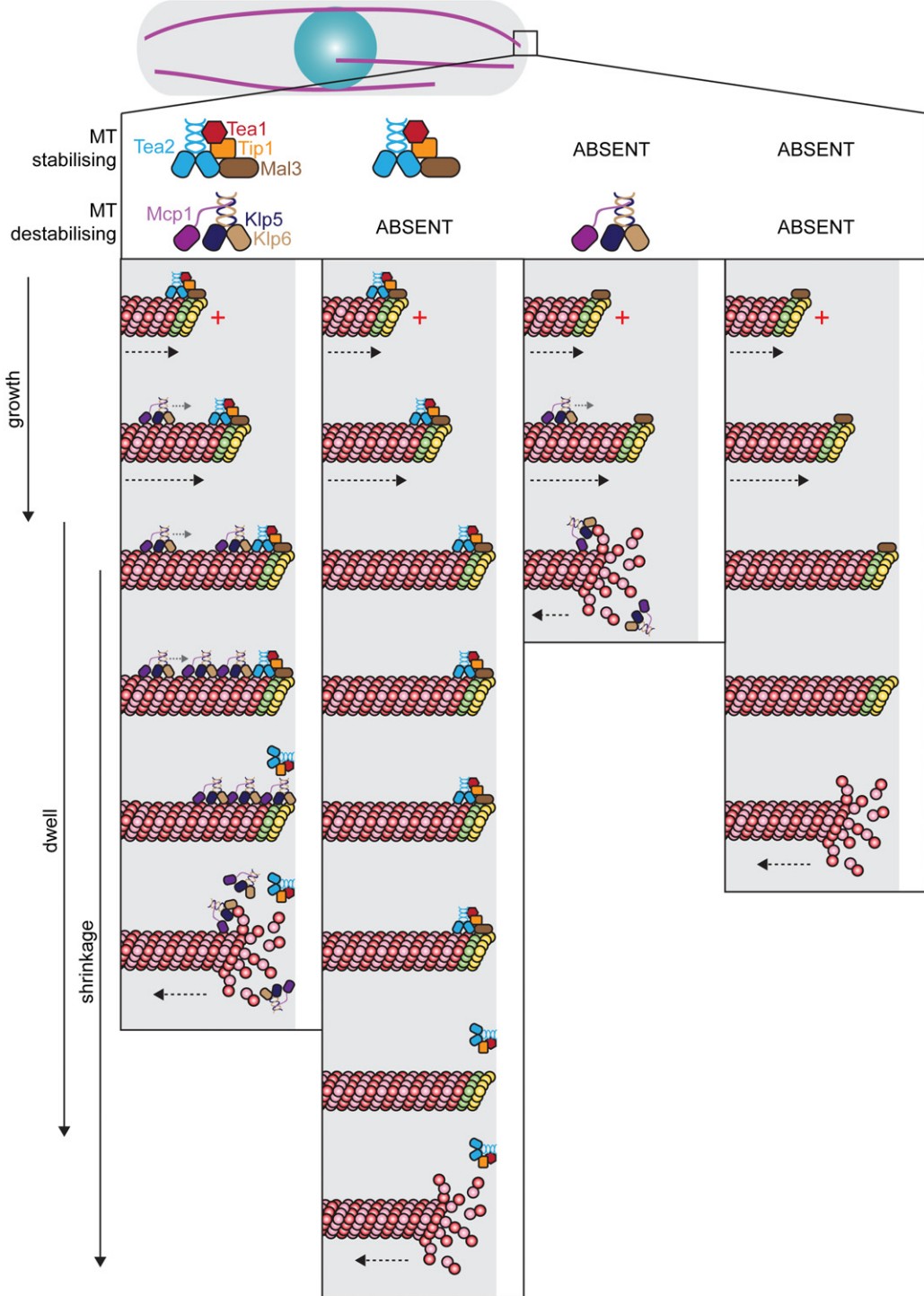

**Figure 5.   Destabilising Kinesin-8 accumulates behind the stabilising Tea2 kinesin complex to ensure microtubule disassembly at the cell end.**

The MT-stabilising Tea2 complex (Tea2/Tip1/Mal3) binds to MT plus ends containing GTP-β-tubulin (green and yellow subunits) as they grow towards the cell end. The MT-destabilising Kinesin-8 (Klp5/Klp6/Mcp1) complex walks along the MT lattice and accumulates behind the Tea2 complex, particularly as the MT dwells at the cell end, but is prevented from accessing the plus end by the presence of Tea2. The Kinesin-8 complex may then either displace the Tea2 complex from the plus end, triggering MT catastrophe, or initiate MT depolymerisation resulting in the loss of the Tea2 complex and the GTP cap. An alternative, but not necessarily exclusive, possibility is that displacement of Tea2 from the MT plus end initiates Kinesin-8-mediated MT disassembly. In the absence of the MT-destabilising Klp5/Klp6/Mcp1 complex (second column), MTs dwell for longer at the cell end and the Tea2 complex remains at the plus end during this period. Klp5/Klp6/Mcp1 is therefore essential for regulating timely dissociation of the Tea2 complex from plus ends. In the absence of Tea2 (third column), Kinesin-8 accumulates at, and has unrestricted access to, the plus end resulting in frequent MT catastrophe before reaching the cell end. When both stabilising and destabilising kinesin complexes are absent (fourth column), MTs reach and dwell at the cell end and may undergo catastrophe as a consequence of compressive force encountered at the cell end.

## Materials and Methods

### Cell culture and strain construction

Media and the growth and maintenance of strains were as described previously [45]. All experiments were performed at 29.5°C unless stated. A full list of strains and oligonucleotides used in this study can be found in Appendix Tables S1 and S2, respectively.

Deletion of *mcp1* with *natMX6*, *hphMX6* and *kanMX6*, C-terminal tagging of *mcp1* with *GFP*, and C-terminal tagging of *klp5* and *klp6* with *mNeonGreen* were carried out by two-step PCR gene targeting as described previously [46]. The *mcp1*, *klp5* and *klp6* genes were also C-terminally tagged with RFP, mCherry or TdTomato. However, the fluorescence intensity of the resultant strains was too low and bleached too quickly to enable time-lapse live-cell imaging. N-terminally tagged *lys1::nmt1-GFP-mcp1* was constructed by amplifying the *mcp1* ORF flanked by *attB1* sites. The resulting PCR product was cloned into *pDONR221*™ and then shuttled to *pLYS1U-HFG1c* using Gateway LR clonase II enzyme mix (Invitrogen, USA). Following linearisation with NruI, this construct was integrated into *S. pombe* strains at the *lys1* locus and confirmed with PCR.

### Live-cell fluorescence microscopy

Live-cell analyses were performed in imaging chambers (CoverWell PCI-2.5, Grace Bio-Labs) filled with 1 ml of 1% agarose in minimal medium and sealed with a 22 × 22 mm glass coverslip.

Figures 1B, EV1A and B, 2A and E, and EV3B use a single-camera Andor spinning disc confocal system comprising of a Nikon Ti-Eclipse microscope base equipped with a Plan Apo Vc 100×/1.40 N.A. objective lens, a Yokogawa CSU-XI confocal unit, an Andor iXon Ultra EMCCD camera with 2× magnification adapter and Andor iQ3 software. 488 and 561 nm solid-state diode (power 50/100 mW) lasers were used for excitation. All images were processed and analysed using ImageJ (Fiji Life-Line version). Temperature was maintained at 29.5°C by a cage incubator controlled by OKO-touch (OKOLAB, Italy).

Figures 1C–F, EV2F and G, 2B–D and F, EV3A and C–E, 3, 4, EV4 and EV5 are generated using an Andor TuCam spinning disc confocal system comprising of a Nikon Ti-Eclipse microscope base equipped with a 100×/1.45 NA objective lens (Nikon CFI Plan Apo Lambda), a Nikon PFS (perfect focus system), a Yokogawa CSU-X confocal unit, and 2 Andor iXon Ultra EMCCD cameras with 2× magnification adapter and Andor TuCam two camera imaging adapter (beam splitter: Semrock DiR561). The final pixel size with 100× lens is 69 nm/pixel (measured). The light source is an Andor ILE laser unit equipped with solid-state 50 mW lasers (488 and 561 nm). Temperature was maintained at 29.5°C by an OKOLAB hated enclosure.

### Registration of images in the TuCam system

Registration of the two channels recorded simultaneously by two separate cameras was performed after acquisition using a reference image and ImageJ plugins for both image registration and transformation. Before every experiment, three 2-channel reference images of 0.5 μm beads (calibration slide, Tetraspec Molecular Probes) in both the mNeonGreen/GFP and TdTomato/mCherry channels were captured. Following the experiment, reference images were again collected to compare for changes in calibration during the imaging

session. Next, the channels in one of the reference images were registered using similarity transformation in the ImageJ plugin *RVSS* (Register Virtual Stack Slices) which performs registration using translation, rotation and isotropic scaling. This file was further tested on the remaining 2-channel reference images using the ImageJ plugin *TVSS* (Transform Virtual Stack Slices) and the transformation file saved for further use. Registered images were inspected to confirm correct alignment of both channels across the whole field of view. Finally, the transformation file was used to register the channels of the corresponding batch of experimental data.

### Kymograph generation and analysis

Kymographs were produced from data generated on the TuCam system. Mid-log phase cells were imaged for both green (mNeon-Green and GFP) and red (TdTomato and mCherry) tags simultaneously with 150 ms exposure every 200 ms for 2 min using 6% laser power for both channels. PFS was set to ON for all time points. Camera noise was suppressed by filtering all images using the Mean Filter in ImageJ set to 1 × 1. A single focal plane was used to increase temporal resolution. Only MTs that remained in this plane throughout image acquisition were used to generate kymographs with the ImageJ plugin *KymoResliceWide* set to generate average maximum intensities for each frame perpendicular to a manually drawn line 10 pixels wide with spline fit.

Kymographs were analysed for fluorescence levels at the MT plus end relative to MT length using a manually drawn line. MT shrinkage was defined as the first frame in which unambiguous MT shrinkage was first observed. MT growth speed was measured by calculating the gradient of mCherry through time during the portion of the kymograph where MTs were not in contact with the cortex. Klp5/Klp6 speed was measured by calculating the gradient of lines tracking mNeonGreen puncta on MT lattices. For the analyses presented in Figs 3, 4, EV4 and EV5 relating to maximal intensity pixel values, both channels from kymographs were saved as text images giving the intensity of every pixel for both channels. The pixel with maximal intensity was then determined for each channel at every time point and its position and fluorescence level recorded. The numbered plots that correspond to the numbered 20 pixels (4 s) wide segments of the kymographs show averaged fluorescence levels for each channel (relative to 100% maximum intensity for each trace) and position. These plots were generated using the *Gel Analyzer* tool in ImageJ.

### Relative fluorescence intensity quantification

To compare the levels of the fluorescently tagged proteins in two strain backgrounds, we performed mixing experiments to allow quantification within the same field of view on the TuCam system. Mid-log phase cells were excited for both green (mNeonGreen and GFP) and red (mCherry) tags simultaneously with 150 ms exposure for each of 11 Z sections (0.3 μm apart) every 2 s for 4 min using 6% laser power for both channels. PFS was initialised every five time points. Camera noise was suppressed by filtering all images using the Mean Filter in ImageJ set to 1 × 1, and Z-stacks were flattened by maximum intensity projection. Fluorescence levels were quantified using a 5 × 5 pixel oval region of interest and determining the mean value. For plus end values, this signal had to be associated with the end of an iMT and was recorded in the frame

immediately prior to MT shrinkage; for nuclear signals, measurements were taken in the first frame. Background subtraction was performed against the cytoplasmic signal in all instances. Mitotic cells were excluded from the analysis.

### iMT Dwell time analysis

Cells expressing *GFP-atb2* were grown until mid-log phase and then MTs imaged live using the single-camera Andor spinning disc confocal system. Eight z-stacks (0.5 μm apart) were taken with exposure times of 100 ms every 5 s for 10 min for GFP. These time-lapse series were then flattened in the Z dimension by maximum intensity projection in ImageJ. Dwell time was defined as the length of time the iMT plus end remained in the final 1.1 μm of a cell, with the cell perimeter defined by background GFP fluorescence. 1.1 μm is the mean length (plus one standard deviation) for the curved end of control cells and accounts for 84% of shrinkage events. Dwell times were measured for the full 10 min of each movie. ~100 dwell times were measured for each condition.

### Imaging of plus end proteins relative to iMT length

For Figs EV1A and B, and 2A, mid-log phase cells expressing fluorescently tagged Klp5, Klp6, Tea2 or Mcp1, together with fluorescently tagged MTs, were imaged live on the single-camera Andor system. Eight z-stacks (0.5 μm apart) were taken with green fluorophores (GFP and mNeonGreen) and red fluorophores (mCherry) excited for 200 ms every 4.8 s for no more than 5 min. Movement of Mcp1 puncta over time (Fig EV1B) was tracked using the ImageJ plugin *MTrackJ*. Klp5/Klp6 and Tea2 levels at plus ends relative to MT length (Figs 2A and EV3D) were analysed manually with mNeonGreen and GFP levels background substituted and normalised against the mCherry mean intensity of iMTs. iMT length was measured by using the line tool in ImageJ with the anti-parallel MT bundle used to define the minus end. Measurements were performed over three independent experiments.

### Measurement of the pre-anaphase mitotic index

Mid-log phase *fta3-GFP sid4-TdTomato* or *cdc13-GFP sid4-TdTomato* cells were grown at 30°C and, following fixation in 3.7% formaldehyde for 10 min, mounted in medium containing DAPI to label DNA. Stacks of 18 z sections (0.2 μm apart) were taken on a wide field microscope system using a Nikon TE-2000 base with a 100×/1.49 N.A. objective lens equipped with a Photometrics Coolsnap-HQ2 liquid cooled CCD camera and analysed using MetaMorph (version 7.5.2.0, MAG, Biosystems Software). The percentage of cells with Cdc13-GFP on the spindle pole bodies and mitotic spindle or the percentage of cells with kinetochores between separated poles prior to anaphase was determined. Exposure times of 1 s were used for GFP and TdTomato and 0.25 s for DAPI. For each experiment, at least 500 cells were counted, and each experiment was conducted three times.

### Analysis of cell curvature

Since polarity defects are emphasised in longer cells, *cdc25-22* cells were grown at 28°C until mid-log phase at which point they were shifted to 35.5°C for 6 h to arrest at G2/M and then imaged live on

the Nikon wide field system. Eighteen z-stacks (0.2 μm apart) were taken with a 100 ms exposure time for bright field. Each experiment analysed at least 280 cells, was repeated three times and excluded cells that were less than 14 μm long. The z-stack containing the middle of the cell was used for analysis, whereby a line was drawn manually between the two cell ends in the middle of the cell using the line tool in ImageJ. The length of this line (L, length) was then compared to that of a straight line between the cell ends (E, Euclidean distance) using the measure function of ImageJ. The ratio between these values, converted to a percentage, gives a measure of the curvature of each cell. The distribution of cell lengths was unaltered between backgrounds.

### Biochemistry

Cells were lysed in buffer containing 50 mM HEPES (pH 7.6), 75 mM KCl, 1 mM $MgCl_2$, 1 mM EGTA, 0.1% Triton X-100, 1 mM DTT, 1 mM PMSF and complete EDTA-free protease inhibitor cocktail (Roche). Protein concentration was quantified by Bradford assay (Bio-Rad, USA). For co-immunoprecipitation, 2 mg protein extract was incubated with 5 μl (~5 μg) rabbit αGFP (Immune Systems) (Immune sera, I) or 2.5 μl (~5 μg) normal rabbit serum (preimmune sera, PI) for 30 min and subsequently with 20 μl protein A-sepharose beads (GE Healthcare, USA) for 45 min. Following centrifugation at 4°C, beads were washed three times in lysis buffer before being heated at 100°C in SDS sample buffer for 6 min.

Precipitates using immune (I) or pre-immune (PI) sera and whole cell extracts (WCE) were migrated using SDS–PAGE and transferred to nitrocellulose membrane. GFP was detected using anti-GFP sheep polyclonal (1/5,000, Kevin Hardwick) and HRP-conjugated anti-sheep secondary antibody (1/10,000, Jackson ImmunoResearch, USA). Myc was detected using anti-Myc mouse monoclonal (1/500, Abcam, UK) and HRP-conjugated anti-mouse secondary antibody (1/10,000, GE Healthcare, USA). Tubulin was detected by anti-TAT1 mouse monoclonal (1/1,000, Keith Gull) and HRP-conjugated anti-mouse secondary antibody (1/10,000, GE Healthcare, USA). ECL detection was performed using Amersham ECL system (GE Healthcare).

### Statistics

Since some data sets were not normally distributed, as ascertained by the Shapiro–Wilk test, the nonparametric two-sample Kolmogorov–Smirnov test was used throughout to determine the probability that two data sets come from different distributions. Text files were manipulated in Excel (version 15.33, Microsoft), and data were analysed by R (version 3.3.3, R Core Team) and RStudio (version 1.0.136, RStudio, Inc).

**Expanded View** for this article is available online.

### Acknowledgements

The authors are grateful to Kevin Hardwick and Keith Gull for their gifts of antibodies and to Fred Chang, Ken Sawin, Takashi Toda, Xiangwei He, Mitsuhiro Yanagida, Jean-Paul Javerzat and Paul Nurse for providing fission yeast strains. This work was funded by a Medical Research Council UK programme grant (MR/K001000/1) to J.B.A.M. Wellcome Trust Senior Investigator Awards to R.A.C. (103895/Z/14/Z) and M.K.B. (WT101885MA), an ERC Advanced Grant

(Actomyosin Ring-ERC-2014-ADG No. 671083) to M.K.B. and a Royal Society Wolfson Merit Award (WM130042) to M.K.B. J.C.M. was funded by a University of Warwick, Institute of Advanced Study, Global Research Fellowship. L.J.M. was funded by a University of Warwick, Chancellor's Scholarship and enrolled in their Medical Research Council UK funded Doctoral Training Partnership.

## Author contributions

JCM, LJM, TCL and JBAM constructed strains. JCM, LJM and AK conducted the experiments and analysed the data. JCM produced the figures and JCM and JBAM wrote the manuscript with contributions from LJM, MKB and RAC. JBAM conceived and supervised the project.

## Conflict of interest

The authors declare that they have no conflict of interest.

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
