## [Review Process File · EMBO Reports]

Opposing kinesin complexes queue at plus tips to ensure microtubule catastrophe at cell ends

John C. Meadows, Liam J. Messin, Anton Kamnev, Theresa C. Lancaster, Mohan K. Balasubramanian, Robert A. Cross and Jonathan B.A. Millar

Review timeline:	Submission date:	28th Mar 18
	Editorial Decision:	18th May 18
	Revision received:	6th Jul 18
	Editorial Decision:	8th Aug 18
	Revision received:	21st Aug 18
	Accepted:	23rd Aug 18

Transaction Report:

1st Editorial Decision

18th May 18

Thank you for the submission of your research manuscript to our journal. I apologize again for the delay in handling your manuscript but as you know, we have only recently received the full set of referee reports and I have further discussed these with the referees.

As you will see, the referees acknowledge that the findings are potentially interesting but they also indicate that substantial further work will be required to substantiate the findings. The referees point out that further data are needed to strengthen the conclusion that there is competition between the kinesin-7 and kinesin-8 complexes. The role of Mcp1, its interaction with Klp5/6 and its impact on MT destabilization needs further support. In general, more quantification and a higher number of observations are needed as pointed out by referee 1 and 2. Moreover, referee 3 remarks that some of the current data on Mcp1/Klp5/6 are in contrast to earlier work, such as the speed of Klp5/6 or that enhancement of MT catastrophe occurs at MTs that are not near the cell tip. Please discuss these discrepancies in the revised version.

I note that referee 2 points out that the data appear too preliminary for publication and I also note that quite some work will be required to substantiate the findings. Yet, given the potential interest of your findings and the novel concept, and the overall constructive comments, I would like to give you the chance to revise your manuscript with the understanding that the referee concerns (as detailed above and in their reports) must be fully addressed and their suggestions taken on board. Please address all referee concerns in a complete point-by-point response. Acceptance of the manuscript will depend on a positive outcome of a second round of review. It is EMBO reports policy to allow a single round of revision only and acceptance or rejection of the manuscript will therefore depend on the completeness of your responses included in the next, final version of the manuscript.

REFeree COMMENTS

Referee #1:

This interesting report describes the relationship between kinesin-7 and kinesin-8 complexes to control microtubule (MT) catastrophes in interphase fission yeast cells. In these cells, MTs normally grow until they reach cell poles, where they deliver cargoes and push against the cell end before undergoing catastrophe. Both kinesin-7 (Tea2 complex) and kinesin-8 (Klp5/6) are known to associate with MT plus ends and kinesin-7 has been well described to protect MTs against premature catastrophe in the cytosol.

In this report, the authors first show that Mcp1 is part of the kinesin-8 complex. This is largely convincing, though there are a few quantifications missing to support the text:

- The authors claim that the intensity of Mcp1 increases at plus tips of iMTs, but it is not clear where this data is shown. Fig EV1A, which is referred to is not quantified.
- It is also not clear why *mcp1Δ* does not show polarization defects like *klp5/6Δ* (in *cdc25-22* background). Maybe the authors can speculate on this.

They then focus their investigation on the links between kinesin-7 and kinesin-8 in regulating MT catastrophe. They show that kinesin-7 and kinesin-8 have opposing effects, protecting vs. promoting catastrophes, respectively. The genetic data on MT dwell times supports this well and, interestingly, they find that kinesin-8 accumulates behind kinesin-7 on the MT until the MT dwells at the cell end and kinesin-8 catches up on kinesin-7. They interpret these data as a competition between the two kinesin complexes, where MT-stabilising Tea2/Tip1/Mal3 kinesin-7 complex binds ahead of the MT-destabilising Klp5/Klp6/Mcp1 kinesin-8 complex and restricts its access to the iMT plus tip until the MT reaches the cell end. This is an interesting idea and nice model, but this is not completely shown. The reduced levels of Klp5/6 at MT tips in *tea2Δ* cells could be because Tea2 no longer competes out Klp5/6 from the MT tips, as suggested by the authors. It could also be due to the fact that MTs are shorter and thus Klp5/6 has less time to accumulate at tips. Two experiments could be done to test this point more thoroughly. First, the authors could restrict their analysis of Klp5/6 levels to MTs of same length to disentangle effects due to MT growth history from direct effects of Tea2. Second, they should test whether Klp5/6 indeed occupies a more distal position on MT ends in absence or Tea2, as predicted by a competition situation. This could be done by comparing the relative position of Klp5/6 with that of Atb2 in WT vs. *tea2Δ* cells.

Finally, in absence of *mcp1*, they claim that Klp5/6 remain behind Tea2 for a longer length of time. The authors should quantify this increased length of time. At present only a few examples are shown.

In summary, I find the study well conducted, but lacking some important quantification. When these are done, I would be happy to support publication of this interesting report.

Minor comments:

- Pay attention to grammar in first sentence of abstract!
- In panel of figure EV1A, it looks like Klp5 stays at the microtubule end even after catastrophe. I do not understand this relative to data shown in Fig 1C.
- In the quantifications of Klp5/6 intensity at MT plus tips, the value given for control cells is not the same in different experiments. For instance it is about 700 in Fig EV3D, but less than 400 in Fig EV3C, and then about 600 in Fig 1F. What is the reason for this variability, which raises concerns about the ability to compare intensities with mutant cells.

Referee #2:

In the manuscript "Opposing kinesin complexes queue at plus tips to ensure microtubule catastrophe at cell ends" the authors investigate the accumulation of microtubule (MT) associated, motor driven, complexes at MT plus tips in interphase fission yeast. This is an important question in the fission

yeast field in which plus tip tracking complexes have demonstrated physiological relevance during the cell cycle, in particular during the establishment of polarity. The authors investigated the interplay between two functionally distinct motor driven complexes that are involved in the regulation of MT dynamics. The broader relevance of this question still needs to be established, but it is likely that motor driven complexes play important roles in other cellular systems than fission yeast. The data presented in the manuscript provides preliminary evidence for a possibly new type of interaction between motor-driven complexes. However, strong evidence for several of the strong claims in the manuscript is lacking. Overall the addressed research question as well as the overall results are novel. Taken together the manuscript idea and topic might be suitable for publication, however in the current presentation of results and due to insufficient evidence publication cannot be recommended.

Questions for referees:

1. Does this manuscript report a single key finding? YES. The authors find that interphase microtubule dynamics in fission yeast is influenced by interacting motor-driven protein complexes.
2. Is the reported work of significance, or does it describe a confirmatory finding or one that has already been documented using other methods or in other organisms etc? YES, the reported work is of significance.
3. Is it of general interest to the molecular biology community? NO. This work is of interest to the fission yeast community, in which MT-cortex interactions play a specific role in the cell cycle.
4. Is the single major finding robustly documented using independent lines of experimental evidence (YES), or is it really just a preliminary report requiring significant further data to become convincing, and thus more suited to a longer format article (NO)? NO. Indeed the authors use independent lines of evidence, however the quality of evidence for their claims is poor and difficult to access for the reader.

Major comments:

a) Claims about Mcp1 and klp5/6 interaction:

The authors use a cartoon, Fig 1A, to present a result and put the actual data in the supporting Figure, reducing the text accessibility. Based on the claim that Mcp1 intensity increases at plus ends as MTs grow, the reader would expect that the actual supportive data shows precisely this. However, instead Fig. EV1A shows a static MT close to the cell edge with a static Mcp1 puncta. The next claim that binding of Mcp1 requires motor activity is presented in a similar unclear way. While the authors indicate later in the manuscript that klp5/6 moves at speeds of 134 nm/s (Fig 1D), Fig. EV1B plots distance of Mcp1 puncta travelled (n=5) vs. time. Performing the calculation shows that Mcp1 puncta travel at 37.5 nm/s (blue curve). This is slower than what is presented in Fig 1D, where they compare the MT growth speed and the motor velocity in the presence of Mcp1 (n=44). The difference in the number of observations seems also large. Why are only very few (n=5) observations of Mcp1 movement reported?

b) Evidence that Mcp1 is required for MT destabilizing function of klp5/6:

The authors investigate the velocities of MT growth and motor speed in Fig 1C-E and appear to conclude from this that Mcp1 is required for the destabilizing function of klp5/6 at cell ends. However, evidence for this claim is not presented in these figures, but in Figure 1B. This should be rephrased. The title of this section furthermore refers to catastrophe activity, which is much more difficult to assess than measuring dwell times at cell ends. To report catastrophe activity it would be necessary to investigate catastrophe frequencies. I assume this is beyond the scope of this work. It needs hundreds of observations to map this quantity accurately and therefore the claim cannot be kept up in the current manuscript. Also, it appears from Fig. 1C that there are multiple MTs in the kymograph, which are sometimes sliding backward (judging from the MT speckles in the bottom panel). It thus seems difficult to draw strong conclusions from the quantifications shown in Fig. 1E.

c) Evidence for dwell-time dependence of accumulations:

In this paragraph (and elsewhere) the authors also claim that Klp5/6/Mcp1 complex accumulates on MTs in a dwell time dependent manner. There are no data supporting this claim in the current

manuscript. Evidence supporting such a claim has been presented in the kip3 paper by Varga et al 2009, see this reference for studying dwell-time dependencies.

d) Tea2-dependency on Klp5/Klp6/Mpc1:

The authors report that they were 'surprised' to see that tea2 localization is unaffected by klp5/6 and Mpc1. It remains elusive to the reader why this should be surprising. It is well known that tea2/tip1/mal3 constitutes a tip-tracking complex, in which no other interactions are necessary, see Bieling et al 2007 and many others. The authors continue their discussion and conclude that, by seeing no influence of klp5/6/Mpc1 on tea2, 'the Klp5/Klp6/Mpc1 complex is required for timely dissociation of Tea2 complex from plus tips.' How the authors arrive at his conclusion is not clear. The removal of Tea2 is presumable simply triggered by a transition to MT shrinkage.

e) Statistics and significance of data:

The number of observations the authors present in their work is quite low. I would like to encourage the authors to increase the quality of their data by improving their statistics.

Minor comments:

- Abstract: typo 'catastrophase'; misuse of the word 'accumulation', please clarify: proteins can accumulate, but not events. In the case of event accumulation it is necessary to talk about frequencies or probabilities.
- Overstatement of results (in introduction): 'reveal a novel model of length control' is not correct. The authors should rather say that their observations lead them to propose a mechanism. The experimental evidence is too poor to make strong claims.
- Poor readability of too long sentences.
- Speculations about mechanisms shall be kept to a minimum and if it is not possible to avoid them then a statement should be clearly recognizable as speculative.
- Lack of reference: 'in the absence of Tea2/Tip1/Mal3 complex, but not its cargo, ...' it remains unclear what the cargo is.

Referee #3:

In this manuscript, Meadows et al study the length regulation of interphase microtubules in fission yeast using live-cell imaging, genetics, and biochemistry. They identify the poorly studied protein Mpc1 as a component of the kinesin-8 Klp5/6 complex that promotes microtubule catastrophe. They propose a new competition model for microtubule length regulation, in which the Tea2/Tip1/Teal1 complex at microtubule plus ends stabilizes the microtubules, until the microtubules reach the cell ends and their growth slows, allowing the complex of Klp5/Klp6/Mpc1 to catch up and push the stabilizing complex off, promoting catastrophe. The high-time-resolution imaging is well applied to address this interesting problem and the authors' results are significant. The results demonstrate the interplay of multiple factors in controlling microtubule dynamics and length. I recommend publication of the paper after the authors consider the following suggestions.

1. I suggest that the authors avoid using the term "antenna model" to describe the model originally proposed by Joe Howard in which processive motors collect in greater numbers along longer MTs, causing more to reach the tips. This mechanism is nothing like an antenna, which works based on an electromagnetic resonance that does not occur for motor proteins, making it a misleading analogy. A possible other name might be "length-dependent catastrophe model."

2. The paper would benefit from more extensive citations to the literature and discussion of the relationship to previous work to place the authors' results in context. I suggest further discussion in two areas:

a. The introduction and discussion focus more on Klp5/6 and its function, and would benefit from further discussion of the Tea2/Mal3/Tip1 complex and its molecular mechanisms. Beinhauer et al 1997 should be cited for the initial characterization of Mal3 in fission yeast. The authors should also cite the work of Browning on the interactions of Tea2, Mal3, and Tip1 (Browning, Hackney, and

Nurse 2003; Browning and Hackney 2005). Discussion of this work should mention that Mal3 appears important for the loading and/or processivity of Tea2. This would also allow the authors to discuss possible molecular mechanisms of MT stabilization by the Tea2/Mal3/Tip1 complex, which could occur from Tip1, from Mal3 (Sandblad et al 2006, des Georges et al 2008, von Loeffelholz 2017), or from activity of Tea2 directly (although I am not aware of evidence for this).

b. Some related work on Klp5/6 should be cited and discussed. Erent et al 2012 found much slower speeds of Klp5/6 movement toward MT plus ends in vitro, which suggested that Klp5/6 could not track growing MT plus ends in cells; some discussion of why the measured motor speeds are so different from the current manuscript's work would be helpful. Unsworth et al 2008 measured changes in interphase MT dynamics in Klp5/6 deletion strains. Tang et al 2015 also discussed PP1 recruitment to the kinetochore by Klp5/6 for checkpoint silencing, and should be cited along with Meadows et al 2011. Gergely et al 2016 found evidence that Klp5 homodimerization may be relevant during mitosis, suggesting that the manuscript's claim that Klp5/6 is an obligate heterodimer should be updated. The authors do cite Tischer et al 2009 in the context of force-dependent catastrophe, but Tischer et al also found direct evidence for a role of Klp5/6 in length-dependent catastrophe of interphase pombe MTs, which should be discussed. It's not clear to me how to reconcile this manuscript's finding that Klp5/6/Mcp1 is typically behind the MT plus end until the MT reaches the cell tip with Tischer et al's result that length-dependent enhancement of MT catastrophe occurs in the presence of Klp5/6, for MTs that are not near the cell tip. The authors should discuss this.

3. In Fig. 1C, the kymograph looks as if there are some shorter MTs, not long enough to reach the top of the figure, that are growing and shortening at about the same speed, not showing the rapid shortening behavior that is characteristic of dynamic instability for single MTs. Is this a correct interpretation of the figure, and does it imply that bundling is having a big effect on MT behavior that should be noted?

4. In Fig. EV1, it looks as if Klp5GFP stays associated with the tip of a shortening MT for quite a while and at constant brightness. This appears different from behavior previously reported for Kip3. Can the authors comment on this?

5. In Fig. EV1C, is the gel lane WCE a Coomassie stained gel or another immunoblot?

6. In Fig. 5 it would be useful to have labels at the top of each column, so one's eye could comprehend more clearly what is spelled out in the legend.

1st Revision - authors' response

6th Jul 18

Referee #1:

This interesting report describes the relationship between kinesin-7 and kinesin-8 complexes to control microtubule (MT) catastrophes in interphase fission yeast cells. In these cells, MTs normally grow until they reach cell poles, where they deliver cargoes and push against the cell end before undergoing catastrophe. Both kinesin-7 (Tea2 complex) and kinesin-8 (Klp5/6) are known to associate with MT plus ends and kinesin-7 has been well described to protect MTs against premature catastrophe in the cytosol. In this report, the authors first show that Mcp1 is part of the kinesin-8 complex. This is largely convincing, though there are a few quantifications missing to support the text:

The authors claim that the intensity of Mcp1 increases at plus tips of iMTs, but it is not clear where this data is shown. Fig EV1A, which is referred to is not quantified.

Fig EV1A now shows quantified Mcp1-GFP intensity data over time for 20 movies of iMTs. This clearly illustrates that Mcp1 accumulates at MT plus ends prior to MT shrinkage, just like Klp5/6. We have also replaced the representative movie of Mcp1-GFP with a movie showing phases of MT growth, dwell and shrinkage rather than just dwell and shrinkage as shown previously. This was an avoidable mistake on our part.

It is also not clear why *mcp1Δ* does not show polarization defects like *klp5/6Δ* (in *cdc25-22* background). Maybe the authors can speculate on this.

We previously showed that Klp5/6 can bind Dis2 (type-1-phosphatase, PP1) to aid silencing of the spindle assembly checkpoint (Meadows et al., *Developmental Cell*, 2011). Our preliminary data indicates that *klp5^{PP1mut} klp6^{PP1mut}* double mutants are defective in interphase polarity control in *cdc25-22* cells. This suggests that, even in the absence of Mcp1, PP1 bound to Klp5/6 localises at MT plus ends to maintain cell polarity. However, as this is not the main thrust of this manuscript we have not included this data but merely raise it as speculation. Interestingly, PP1 also binds Tea4, a cargo of the Tea2/Tip1/Mal3 complex, to influence cell polarity in fission yeast (Alvarez-Tabarés et al., *J Cell Science*, 2007; Kokkoris et al., *J. Cell Science*, 2014).

They then focus their investigation on the links between kinesin-7 and kinesin-8 in regulating MT catastrophe. They show that kinesin-7 and kinesin-8 have opposing effects, protecting vs. promoting catastrophes, respectively. The genetic data on MT dwell times supports this well and, interestingly, they find that kinesin-8 accumulates behind kinesin-7 on the MT until the MT dwells at the cell end and kinesin-8 catches up on kinesin-7. They interpret these data as a competition between the two kinesin complexes, where MT-stabilising Tea2/Tip1/Mal3 kinesin-7 complex binds ahead of the MT-destabilising Klp5/Klp6/Mcp1 kinesin-8 complex and restricts its access to the iMT plus tip until the MT reaches the cell end. This is an interesting idea and nice model, but this is not completely shown. The reduced levels of Klp5/6 at MT tips in *tea2Δ* cells could be because Tea2 no longer competes out Klp5/6 from the MT tips, as suggested by the authors. It could also be due to the fact that MTs are shorter and thus Klp5/6 has less time to accumulate at tips. Two experiments could be done to test this point more thoroughly. First, the authors could restrict their analysis of Klp5/6 levels to MTs of same length to disentangle effects due to MT growth history from direct effects of Tea2.

We do not suggest that Tea2 necessarily outcompetes Klp5/6 from iMT plus ends. It could be that Tea2 removal is a consequence of iMT shrinkage *via* Klp5/6 accumulation/position. The reviewer is quite correct, however, to point out that the question of whether levels of Klp5/6 are altered in the absence of Tea2 as a function of MT length is directly testable. We have now performed this analysis (Fig EV3D) on cells in the same field of view and find that Klp5/6 levels are indistinguishable for MTs that have yet to reach the cell end ($< \sim 5\mu\text{m}$) in either the presence or absence of Tea2. This means that in the absence of Kinesin-7 complex MTs of comparable length are not intrinsically able to bind more or less Kinesin-8. However, when MTs are longer ($> \sim 5\mu\text{m}$), the decreased MT life-time at cell ends in the absence of Kinesin-7 (see dwell time (Fig 2E), location of MT shrinkage (Fig 2F) & distribution of 100 MT lengths (Fig EV3D) \pm Tea2) demonstrates that long-lived MTs bind more Klp5/6. The reviewer is therefore correct that "*MTs are shorter and thus Klp5/6 has less time to accumulate at tips*". We now make this conclusion explicit in the manuscript.

Second, they should test whether Klp5/6 indeed occupies a more distal position on MT ends in absence or Tea2, as predicted by a competition situation. This could be done by comparing the relative position of Klp5/6 with that of Atb2 in WT vs. *tea2Δ* cells.

This suggestion is not mutually exclusive with the reviewer's previous hypothesis. We attempted these experiments but repeatedly found that the fluorescently-tagged MT signal was, unfortunately, not well enough defined (poor local signal-to-noise ratio) to get sufficiently accurate data to determine the exact position of the MT plus end. Presumably the influx/efflux of fluorescently-tagged tubulin dimers around the MT plus tip and the dynamic structure of the MT plus tip itself create a diffuse fluorescence signal that means the precise MT end cannot be defined at this scale.

Finally, in absence of *mcp1*, they claim that Klp5/6 remain behind Tea2 for a longer length of time. The authors should quantify this increased length of time. At present only a few examples are shown.

The difference in the duration of Klp5/6 remaining behind the T-complex is dependent on the increased MT dwell time in the absence of Mcp1 as shown in Fig 1B. We quantified the increased time that Klp5/6 remains behind the T-complex in Figs 4B & 4C (data from 7 control cells and 8 $\Delta mcp1$ cells, respectively). We show 5 examples of Klp5/6 behind T-complex for control cells (1 in Fig 4A & 4 in Fig EV4) and one example in $\Delta mcp1$ cells (Fig EV5). The salient point is that Kinesin-8 is always behind Kinesin-7 until just prior to shrinkage detection. At the resolution of our experiments, Kinesin-8 complex invasion appears coincident with catastrophe and remains behind the T-complex for longer in $\Delta mcp1$ cells because MT life-time is prolonged.

In summary, I find the study well conducted, but lacking some important quantification. When these are done, I would be happy to support publication of this interesting report.

Minor comments:

Pay attention to grammar in first sentence of abstract!

This has now been rectified.

In panel of figure EV1A, it looks like Klp5 stays at the microtubule end even after catastrophe. I do not understand this relative to data shown in Fig 1C.

We thank the reviewer for bringing this to our attention. This discrepancy stems from the fact that Fig EV1A was performed by imaging every 4.8 s rather than by using a dual camera (TuCam) spinning disc microscope which enabled imaging every 200 ms (detailed in the Materials and Methods section). Using the higher temporal resolution, it is apparent that the majority of Klp5/6 disappears from the microtubule plus end just prior to MT shrinkage but new Klp5/6 complexes continue to travel to the depolymerising plus end. This can clearly be seen in Fig EV4 (iv) where a population of Klp5/6 reaches the depolymerising plus end in frame 14. However, to remove any unnecessary confusion, we now present an alternative representative movie in Fig EV1A.

To provide additional clarity we now add quantification of Klp5-GFP accumulation on 20 iMTs to present Klp5 accumulation at iMT plus ends and its removal relative to MT shrinkage (Fig EV1A). This is consistent with data in Figs 1C, 1E, 4A-C, EV4 & EV5.

In the quantifications of Klp5/6 intensity at MT plus tips, the value given for control cells is not the same in different experiments. For instance it is about 700 in Fig EV3D, but less than 400 in Fig EV3C, and then about 600 in Fig 1F. What is the reason for this variability, which raises concerns about the ability to compare intensities with mutant cells.

The intensity of Klp5/6 in any given experiment is given in Arbitrary Units (AU). It is not directly comparable between experiments because imaging conditions (e.g. agar properties, laser power, sample preparation) can change. Indeed, this inherent variability is the very reason why we have performed a series of mixing experiments imaging cells in the same field of view (Figs 1F & EV3C-E). This allows for far more accurate extraction of relative intensities and allows for direct comparisons of fluorescence values.

Referee #2:

In the manuscript "Opposing kinesin complexes queue at plus tips to ensure microtubule catastrophe at cell ends" the authors investigate the accumulation of microtubule (MT) associated, motor driven, complexes at MT plus tips in interphase fission yeast. This is an important question in the fission yeast field in which plus tip tracking complexes have demonstrated physiological relevance during the cell cycle, in particular during the establishment of polarity. The authors investigated the interplay between two functionally distinct motor driven complexes that are involved in the regulation of MT dynamics. The broader relevance of this question still needs to be established, but it is likely that motor driven complexes play important roles in other cellular systems than fission yeast. The data presented in the manuscript provides preliminary evidence for a possibly new type of interaction between motor-driven complexes. However, strong evidence for several of the strong claims in the manuscript is lacking. Overall the addressed research question, as well as the overall results, is novel. Taken together the manuscript idea and topic might be suitable for publication, however in the current presentation of results and due to insufficient evidence publication cannot be recommended.

Questions for referees:

1. Does this manuscript report a single key finding? YES. The authors find that interphase microtubule dynamics in fission yeast is influenced by interacting motor-driven protein complexes.

We thank the referee for this endorsement.

2. Is the reported work of significance, or does it describe a confirmatory finding or one that has already been documented using other methods or in other organisms etc? YES, the reported work is of significance.

We also thank the referee for this endorsement.

3. Is it of general interest to the molecular biology community? NO. This work is of interest to the fission yeast community, in which MT-cortex interactions play a specific role in the cell cycle.

We firmly believe that the primary finding of this manuscript, that microtubule length control is an emergent property that reflects spatially-regulated competition between distinct kinesin complexes at the microtubule plus end will be of interest to all cytoskeleton researchers. Crucially, we expand upon the role of the evolutionarily conserved Kinesin-8 family of MT regulators. Just as work on Kip3 (budding yeast) and KIF18 (mammalian) have contributed to our understanding of how these proteins operate, so too does this work on Klp5 and Klp6 (fission yeast). The recent, excellent paper from Pellman and colleagues (Arellano-Santoyo *et al.*, *Dev Cell*, 2017) dissecting the mechanism whereby Kinesin-8 (in this case Kip3) switches from motor activity to depolymerase clearly demonstrates that valid work can, and should, be conducted across multiple organisms to understand universally conserved properties.

Similarly, we find the idea that MT-cortex interactions are somehow uniquely important in fission yeast somewhat disingenuous - they are required across the eukaryotic kingdom for a range of fundamental biological processes! Does the reviewer dismiss Paul Nurse's groundbreaking work on the Tea system in fission yeast (<https://www.ncbi.nlm.nih.gov/pubmed/?term=Nurse+P+polarity>) as not of general interest to molecular biology community?

4. Is the single major finding robustly documented using independent lines of experimental evidence (YES), or is it really just a preliminary report requiring significant further data to become convincing, and thus more suited to a longer-format article (NO)? NO. Indeed the authors use independent lines of evidence, however the quality of evidence for their claims is poor and difficult to access for the reader.

This view is not held by the other referees. We have chosen the short report format as the manuscript concisely and rigorously demonstrates an important major finding that is directly relevant to those interested in understanding microtubule dynamics.

Major comments:

a) Claims about Mcp1 and klp5/6 interaction:

The authors use a cartoon, Fig 1A, to present a result and put the actual data in the supporting Figure, reducing the text accessibility. Based on the claim that Mcp1 intensity increases at plus ends as MTs grow, the reader would expect that the actual supportive data shows precisely this. However, instead Fig. EV1A shows a static MT close to the cell edge with a static Mcp1 puncta.

We feel that the use of a cartoon in Fig 1A helps to orient the non-specialist reader, especially those from outside the fission yeast community, and in fact increases the accessibility of the manuscript. Our aim is to illustrate the dynamics under investigation during the manuscript and their typical cellular location - we then provide a model in the final figure to describe how these dynamics could be regulated based on the data presented throughout.

The reviewer is quite correct that our original Fig EV1A did not support our findings well. We have improved this Figure by now selecting an example with iMT growth, dwell and shrinkage phases to replace the previous figure which showed a MT already paused at the cell end. This was an avoidable mistake on our part. We have also now quantified Mcp1-GFP intensity data over time for 20 iMTs in the same conditions, illustrating that Mcp1 accumulates at MT plus ends prior to MT shrinkage.

The next claim that binding of Mcp1 requires motor activity is presented in a similar unclear way. While the authors indicate later in the manuscript that klp5/6 moves at speeds of 134 nm/s (Fig 1D), Fig. EV1B plots distance of Mcp1 puncta travelled (n=5) vs. time. Performing the calculation shows that Mcp1 puncta travel at 37.5 nm/s (blue curve). This is slower than what is presented in Fig 1D, where they compare the MT growth speed and the motor velocity in the presence of Mcp1 (n=44). The difference in the number of observations seems also large. Why are only very few (n=5) observations of Mcp1 movement reported?

We have added more data showing that movement of Mcp1-GFP puncta at plus tips of iMTs is dependent on the presence of both Klp5 and Klp6 (Fig EV1B). This strengthens our finding that Mcp1 at the MT plus tip requires the motor activity of Kinesin-8. The N for these observations are relatively low because they corroborate previously data shown in Zheng *et al.*, FEBS Lett, 2014 showing that fluorescently-tagged Mcp1 requires Klp6 to localise to the MT plus tip. The result is clear and shown consistently across 5 different strain backgrounds.

The apparent discrepancy in speed between Klp5/6 and Mcp1 that the referee alludes to can be explained because Fig EV1B does not show the speed of Mcp1 on the MT lattice but rather its movement once bound at polymerising iMT plus end. Mcp1 is a very low abundance protein (Marguerat *et al.*, Cell, 2012). Unfortunately, the intensity of Mcp1-GFP on the MT lattice is not sufficiently high to enable us to derive its motorised speed on the microtubule lattice. Instead, the Mcp1-GFP recorded in Figs EV1A-B is that proportion of the protein that has accumulated on the plus tips of iMTs. As such there is no discrepancy. We apologise for our failure to communicate this clearly and have now modified the Figure Legend and manuscript text to better reflect these analyses.

b) Evidence that Mcp1 is required for MT destabilizing function of klp5/6:

The authors investigate the velocities of MT growth and motor speed in Fig 1C-E and appear to conclude from this that Mcp1 is required for the destabilizing function of klp5/6 at cell ends. However, evidence for this claim is not presented in these figures, but in Figure 1B. This should be rephrased.

We thank the reviewer for pointing out this inaccuracy and have rectified it in the text.

The title of this section furthermore refers to catastrophe activity, which is much more difficult to assess than measuring dwell times at cell ends. To report catastrophe activity it would be necessary to investigate catastrophe frequencies. I assume this is beyond the scope of this work. It needs hundreds of observations to map this quantity accurately and therefore the claim cannot be kept up in the current manuscript.

Again, we thank the reviewer for this helpful comment. We had erroneously conflated catastrophe frequency with MT shrinkage whereas of course some catastrophe events will not produce detectable shrinkage before rescue, so there is not necessarily a 1:1 relationship. We have altered the title of the section to better reflect the data.

Also, it appears from Fig. 1C that there are multiple MTs in the kymograph, which are sometimes sliding backward (judging from the MT speckles in the bottom panel). It thus seems difficult to draw strong conclusions from the quantifications shown in Fig. 1E.

The reviewer is very observant. These kymographs are generated from a single focal plane. The lower panel in Fig 1C shows a single iMT (or iMT bundle). The banding is caused by an unequal incorporation of fluorescence into MTs. This appears to slide backwards due to bending of the iMT caused by interaction of the iMT plus end with the cell cortex. When another MT enters the focal plane, as is the case at ~ 50 s in the top panel and ~ 85 s in the bottom panel of Fig 1C respectively, the intensity of the fluorescence doubles. We are therefore confident that we are observing single (or a single bundle) of iMTs and thus the conclusions drawn from the quantification in Fig 1E. We had neglected to mention this in the Figure Legend and this has now been rectified.

c) Evidence for dwell-time dependence of accumulations:

In this paragraph (and elsewhere) the authors also claim that Klp5/6/Mcp1 complex accumulates on MTs in a dwell time dependent manner. There are no data supporting this claim in the current manuscript.

This assessment is simply not accurate. The dwell times detailed in Fig 1B show that in control cells iMTs dwell in the final 1.1 μm of the cell for, on average, 54.2 s and for significantly longer in the absence of Klp5/6/Mcp1. This MT dwell time is represented in Fig 1E, where there is a clear accumulation of Klp5/6 prior to shrinkage both in the presence and absence of Mcp1. It is also represented clearly in the top kymograph in Fig 1C, as Klp5/6 accumulate as the MT pauses whilst it dwells in the final 1.1 μm of the cell. This MT dwell time dependent accumulation of Klp5/6 can again be seen in Figs EV1A, 2A, EV3C-E, 4, EV4 & EV5. Additionally, we now include data showing that during the time iMT dwell at the cell end, Klp5/6 levels increase at a significantly higher rate compared to shorter, less long-lived MTs (Fig EV3D). In fact, when dwell time is significantly reduced (Δtea2 , Fig 2E) Klp5/6 accumulation is likewise decreased - even on those iMTs that do reach the cell end (Fig EV3D). Comparatively, Tea2 & Tip1 do not accumulate during this dwell phase but rather remain at a constant level until MT shrinkage is detected (Figs 2A-C, 3, 4 & EV5).

Evidence supporting such a claim has been presented in the kip3 paper by Varga et al 2009, see this reference for studying dwell-time dependencies.

We respectfully wonder if the reviewer is confusing an individual protein's residence duration at MT plus ends (as measured by Varga and colleagues *in vitro* on stabilised MTs) with our observation that MT dwell time at the ends of live cells correlates with a sustained period of protein complex accumulation? Dwell time in our manuscript, as defined in Fig 1A, refers to the time an iMT spends at the cell end, not to how long each protein resides at the MT plus end.

d) Tea2-dependency on Klp5/Klp6/Mpc1:

The authors report that they were 'surprised' to see that tea2 localization is unaffected by

kIp5/6 and Mpc1.

Again, that is not an accurate description of what we said in the manuscript. Tea2 localisation is affected the loss of KIp5/6 - namely that the Tea2 complex stays at the plus end of MTs for longer in the absence of the KIp5/6/Mcp1 complex. iMT shrinkage is delayed in $\Delta mcp1$ cells but so too is removal of Tea2. We had expected Tea2 to be removed from MTs coincident with MT-cortex contact to establish and maintain polarity but instead find that its timely removal is instead dependent on KIp5/6 activity either directly or indirectly through MT depolymerisation. We still find this surprising.

It remains elusive to the reader why this should be surprising. It is well known that tea2/tip1/mal3 constitutes a tip-tracking complex, in which no other interactions are necessary, see Bieling et al 2007 and many others.

The reason this is surprising is that if MTs dwell for extended periods then the GTP cap would presumably be lost by hydrolysis of GTP to GDP. Since Mal3 preferentially binds GTP tubulin - one would have thought that the Tea2/Tip1/Mal3 complex would dissociate from MTs that dwell for protracted periods. However, this is plainly not the case.

The authors continue their discussion and conclude that, by seeing no influence of kIp5/6/Mpc1 on tea2,

This is again not what we conclude. Tea2 localisation is affected by the loss of the KIp5/6/Mcp1 complex because Tea2 remains localised at the MT plus end for longer in its absence.

'the KIp5/KIp6/Mpc1 complex is required for timely dissociation of Tea2 complex from plus tips.' How the authors arrive at his conclusion is not clear. The removal of Tea2 is presumably simply triggered by a transition to MT shrinkage.

The removal of Tea2 complex may indeed be triggered by a transition to MT shrinkage. However, both the timely transition to MT shrinkage and the timely removal of the Tea2 complex are dependent on the presence of the KIp5/KIp6/Mcp1 complex. So, we are correct in concluding that the KIp5/KIp6/Mcp1 complex is required for *timely* dissociation of Tea2 complex from plus tips.

e) Statistics and significance of data:

The number of observations the authors present in their work is quite low. I would like to encourage the authors to increase the quality of their data by improving their statistics.

We have increased the number of observations for Figs EV1A-B as per reviewer's comment (a). We also include "source data" for all the figures detailing the number of observations, standard deviations and probability values.

Minor comments:

Abstract: typo 'catastrophase'; misuse of the word 'accumulation', please clarify: proteins can accumulate, but not events. In the case of event accumulation it is necessary to talk about frequencies or probabilities.

We are grateful to the reviewer for highlighting this and have modified the text to ensure that this distinction is made.

Overstatement of results (in introduction): 'reveal a novel model of length control' is not correct. The authors should rather say that their observations lead them to propose a mechanism. The experimental evidence is too poor to make strong claims.

We have altered the statement accordingly.

Poor readability of too long sentences.

We have tried to be as accurate as possible in describing a very complex series of events without compromising readability. We have re-written parts of the manuscript, particularly those that contained longer sentences and altered the text to improve readability without losing accuracy.

Speculations about mechanisms shall be kept to a minimum and if it is not possible to avoid them then a statement should be clearly recognizable as speculative.

We think that it is useful for the audience to read the speculations of the authors as to the mechanisms involved. This is part of the scientific process of publishing. However, we are careful to distinguish our conclusions from speculations.

Lack of reference: 'in the absence of Tea2/Tip1/Mal3 complex, but not its cargo, ...' it remains unclear what the cargo is.

We now clarify that the Tea1, Tea3 and Tea4 proteins are cargoes of the Tea2/Tip1/Mal3 complex.

Referee #3:

In this manuscript, Meadows et al study the length regulation of interphase microtubules in fission yeast using live-cell imaging, genetics, and biochemistry. They identify the poorly studied protein Mcp1 as a component of the kinesin-8 Klp5/6 complex that promotes microtubule catastrophe. They propose a new competition model for microtubule length regulation, in which the Tea2/Tip1/Tea1 complex at microtubule plus ends stabilizes the microtubules, until the microtubules reach the cell ends and their growth slows, allowing the complex of Klp5/Klp6/Mcp1 to catch up and push the stabilizing complex off, promoting catastrophe. The high-time-resolution imaging is well applied to address this interesting problem and the authors' results are significant. The results demonstrate the interplay of multiple factors in controlling microtubule dynamics and length. I recommend publication of the paper after the authors consider the following suggestions.

1. I suggest that the authors avoid using the term "antenna model" to describe the model originally proposed by Joe Howard in which processive motors collect in greater numbers along longer MTs, causing more to reach the tips. This mechanism is nothing like an antenna, which works based on an electromagnetic resonance that does not occur for motor proteins, making it a misleading analogy. A possible other name might be "length-dependent catastrophe model."

The reviewer makes an excellent point. Similarly, insect antennae do not function by accumulating sensory information at the apical tip, however, there is good evidence that the longer the antenna the more sensory information it can capture (Lockey & Willis, *J Exp Biol*, 2015). Regardless, we have altered the wording of the abstract to reflect the referee's comment.

2. The paper would benefit from more extensive citations to the literature and discussion of the relationship to previous work to place the authors' results in context. I suggest further discussion in two areas:

a. The introduction and discussion focus more on Klp5/6 and its function, and would benefit from further discussion of the Tea2/Mal3/Tip1 complex and its molecular mechanisms. Beinhauer et al 1997 should be cited for the initial characterization of Mal3 in fission yeast. The authors should also cite the work of Browning on the interactions of Tea2, Mal3, and Tip1 (Browning, Hackney, and Nurse 2003; Browning and Hackney 2005). Discussion of this work should mention that Mal3 appears important for the loading and/or processivity of Tea2. This would also allow the authors to discuss possible molecular mechanisms of MT stabilization by the Tea2/Mal3/Tip1 complex, which could occur from Tip1, from Mal3 (Sandblad et al 2006, des Georges et al 2008, von Loeffelholz 2017), or from activity of

Tea2 directly (although I am not aware of evidence for this).

We thank the reviewer for their diligence regarding literature citations and have added the suggested references to the manuscript where appropriate.

b. Some related work on Klp5/6 should be cited and discussed. Erent et al 2012 found much slower speeds of Klp5/6 movement toward MT plus ends *in vitro*, which suggested that Klp5/6 could not track growing MT plus ends in cells; some discussion of why the measured motor speeds are so different from the current manuscript's work would be helpful.

There are a myriad of reasons why this might be the case. The speed of Klp5/6 *in vitro* was measured by Erent and colleagues using purified truncated Klp5/6 proteins lacking the C-terminal tail regions, which may influence their processivity. As these proteins were purified from bacteria they may also lack key post-translational modifications. Similarly, the buffer conditions may differ from physiological conditions and the researchers used GMPCPP-stabilised MTs, which may alter dynamics. In short, a direct comparison between *in vitro* and *in vivo* speeds is probably not particularly informative and so we do not make this comparison.

Unsworth et al 2008 measured changes in interphase MT dynamics in Klp5/6 deletion strains.

We now include this reference.

Tang et al 2015 also discussed PP1 recruitment to the kientochore by Klp5/6 for checkpoint silencing, and should be cited along with Meadows et al 2011.

We now include this reference.

Gergely et al 2016 found evidence that Klp5 homodimerization may be relevant during mitosis, suggesting that the manuscript's claim that Klp5/6 is an obligate heterodimer should be updated.

The paper from Gergely and colleagues uses single deletion mutants that could artificially allow for homo-dimerisation of kinesin motors only when one protein is absent. We have constructed mutants in Klp5 and Klp6 kinesin motor heads which prevents the hydrolysis of ATP. These mutants fully mimic single and double deletions in all assays, with the exception of spindle checkpoint silencing, which is a motor-independent function. Importantly, the movement of Klp5 is dependent on the motor activity of Klp6 and vice versa. Indeed, to our knowledge, there are no data that support the idea of functional Klp5/5 or Klp6/6 homodimers during interphase. We therefore respectfully decline to update this. We do, however, take on board the reviewer's comment on the language used and therefore replace "*function as an obligate heterodimer*" with "*operate as a functional heterocomplex in interphase*".

The authors do cite Tischer et al 2009 in the context of force-dependent catastrophe, but Tischer et al also found direct evidence for a role of Klp5/6 in length-dependent catastrophe of interphase pombe MTs, which should be discussed.

We now include this reference.

It's not clear to me how to reconcile this manuscript's finding that Klp5/6/Mcp1 is typically behind the MT plus end until the MT reaches the cell tip with Tischer et al's result that length-dependent enhancement of MT catastrophe occurs in the presence of Klp5/6, for MTs that are not near the cell tip. The authors should discuss this.

The reviewer refers to analyses conducted by Tischer and colleagues whereby cell length was increased *via* HU treatment. It is possible that tubulin becomes limiting in longer cells undergoing cell cycle arrest, increasing the stochastic loss of the stabilising GTP-cap

before MTs reach the cell end. This may cause loss of the Tea2 complex without interaction of the MT with the cell end. Our data show that when Tea2 is removed, MT shrinkage events are distributed more broadly throughout the cell and that this distribution depends on the presence of Klp5/6/Mcp1 (Fig 2F). As such our findings do broadly agree with those of Tischer and colleagues, namely that MT catastrophe is enhanced by the presence of Klp5/6 even away from the cell end. The presence of Tea2 complex ensures this normally occurs (in unperturbed cells) once the iMT reaches the cell end.

3. In Fig. 1C, the kymograph looks as if there are some shorter MTs, not long enough to reach the top of the figure, that are growing and shortening at about the same speed, not showing the rapid shortening behavior that is characteristic of dynamic instability for single MTs. Is this a correct interpretation of the figure, and does it imply that bundling is having a big effect on MT behavior that should be noted?

The reviewer is very observant. These kymographs are generated from a single focal plane. The lower panel in Fig 1C shows a single iMT (or iMT bundle). The banding is caused by an unequal incorporation of fluorescence into MTs. This appears to slide backwards due to bending of the iMT caused by interaction of the iMT plus end with the cell cortex. When another MT enters the focal plane, as is the case at ~ 50 s in the top panel and ~ 85 s in the bottom panel of Fig 1C respectively, the intensity of the fluorescence doubles. We are therefore confident that we are observing single (or a single bundle) of iMTs and thus the conclusions drawn from the quantification in Fig 1E are valid. We had neglected to mention this in the Figure Legend and this has now been rectified.

4. In Fig. EV1, it looks as if Klp5GFP stays associated with the tip of a shortening MT for quite a while and at constant brightness. This appears different from behavior previously reported for Kip3. Can the authors comment on this?

We thank the reviewer for bringing this to our attention. This discrepancy stems from the fact that Fig EV1A was performed by imaging every 4.8 s rather than by using a dual camera (TuCam) spinning disc microscope which enables imaging every 200 ms (detailed in the Materials and Methods section). Using the higher temporal resolution, it is apparent that the majority of Klp5/6 disappears from the microtubule plus end just prior to MT shrinkage but new Klp5/6 complexes continue to travel to the depolymerising plus end after shrinkage. This can clearly be seen in Fig EV4 (iv) where a population of Klp5/6 reaches the depolymerising plus end in frame 14. However, to remove any unnecessary confusion, we now present an alternative representative movie in Fig EV1A.

Also, to provide additional clarity for the reader we now add quantification of Klp5-GFP on 20 iMTs demonstrating protein accumulation and removal relative to MT shrinkage (Fig EV1A). This is consistent with other data in Figs 1C, 1E, 4A-C, EV4 & EV5.

5. In Fig. EV1C, is the gel lane WCE a Coomassie stained gel or another immunoblot?

The WCE is another immunoblot - we have added a note to the Figure Legend clarifying this.

6. In Fig. 5 it would be useful to have labels at the top of each column, so one's eye could comprehend more clearly what is spelled out in the legend.

We have now substituted the description "ABSENT" in place of "-" to improve reader accessibility.

2nd Editorial Decision

8th Aug 18

Thank you for the submission of your revised manuscript to EMBO reports. I apologize again for the delay in handling your manuscript but we have now received the reports from referee 1 and 2 and as you will see, both referees are very positive about the study and support publication in EMBO reports.

Browsing through the manuscript myself, I noticed a few things that we need from the editorial side before we can proceed with the official acceptance of your study.

REFEREE COMMENTS

Referee #1:

The authors have addressed my comments adequately.

Referee #2:

The authors have addressed all the points raised by the referees. Improvements include additional references, "source data" for all the figures, clarification of the text as well as the figures. The complex spatio-temporal dynamics is well represented in the revised manuscript and the readability of the manuscript is very good. The revised manuscript is certainly suitable for publication in EMBO reports.

Minor notes

p. 7 contains a typo "klp6/klp6"

Corresponding Author Name: John Meadows & Jonathan Millar

Journal Submitted to: EMBO R

Manuscript Number: EMBOR-2018-46196V2